# Strategies to Improve Vaccination among At-Risk Adults and the Elderly in Italy

**DOI:** 10.3390/vaccines8030358

**Published:** 2020-07-04

**Authors:** Giovanna Elisa Calabrò, Alessia Tognetto, Elettra Carini, Silvia Mancinelli, Laura Sarnari, Vittoria Colamesta, Walter Ricciardi, Chiara de Waure

**Affiliations:** 1Section of Hygiene, Department of Life Sciences and Public Health; Università Cattolica del Sacro Cuore, 00168 Rome, Italy; giovannaelisa.calabro@gmail.com (G.E.C.); alessia.tognetto@gmail.com (A.T.); elettra.carini1@gmail.com (E.C.); walter.ricciardi@unicatt.it (W.R.); 2VIHTALI (Value In Health Technology and Academy for Leadership & Innovation), Spin-Off of Università Cattolica del Sacro Cuore, 00168 Rome, Italy; 3Department of Pneumological Sciences, Section of Pneumology, University of Pavia and Fondazione IRCCS Policlinico San Matteo, 27100 Pavia, Italy; silviamancinelli91@gmail.com; 4Regional Health Unit ASUR AV 3, Sanitary District of Macerata, 62100 Macerata, Italy; sarnari13@gmail.com; 5UOC Direzione Sanitaria S. Spirito e Nuovo Regina Margherita, Local Health Unit ASL RM1, 00193 Rome, Italy; 6Fondazione Policlinico Universitario A. Gemelli IRCCS, 00168 Rome, Italy; 7Department of Experimental Medicine, University of Perugia, 06132 Perugia, Italy; chiara.dewaure@unipg.it

**Keywords:** HZV vaccination, influenza vaccination, pneumococcal vaccination, immunization strategies, elderly

## Abstract

The World Health Organization (WHO), the United States (US) Centers for Disease Control and Prevention (CDC), the European Center for Disease Control (ECDC), and the immunization guidelines of many countries issue vaccination recommendations for adults and the elderly. However, the uptake of vaccination in these groups is generally low due to several reasons. The present study aimed to identify strategies implemented in Italy in unconventional settings to promote vaccination against influenza, pneumococcal, and herpes zoster virus (HZV) infections among these subjects, i.e., the at-risk adult population and the elderly. We conducted a literature review and a survey of experts. The literature search yielded seven strategies; all of these concerned influenza vaccination, while three also addressed pneumococcal and HZV vaccination. The survey of experts identified 15 strategies; 10 regarded influenza vaccination, while four regarded pneumococcal vaccination and one regarded HZV vaccination. Most of the strategies were implemented in hospital clinics and rest homes. Regarding influenza and pneumococcal vaccinations, the target population mainly comprised at-risk adults, while the elderly represented the main target population for HZV vaccination. Our results show that, in Italy, there are initiatives aimed at promoting vaccination in unconventional settings, but further efforts are required to assess their effectiveness and to further extend them.

## 1. Introduction

Vaccinations are effective and cost-effective tools for the prevention of infectious diseases. Nevertheless, a huge number of deaths today are caused by vaccine-preventable diseases (VPDs). According to the latest World Health Organization (WHO) data, vaccination prevents 2–3 million deaths every year [1]. While vaccination can benefit persons of all ages, it is especially important for those at higher risk of infectious diseases and their complications. As the immune system weakens with increasing age, many infectious diseases are more severe and more closely associated with long-term consequences in the elderly than in younger people [2]. Moreover, regardless of age, any health condition that weakens the immune system exposes the person to a greater risk of infectious diseases and their consequences [3]. Therefore, adults presenting with certain characteristics and particular chronic conditions (cardiovascular, respiratory, or metabolic diseases, immunodepression, etc.) are considered “at-risk”. While awareness of childhood vaccination is well established, the importance of vaccination for the elderly and for at-risk individuals is not as well perceived. A recent Eurobarometer survey [4] revealed that, in Europe, only 85% of citizens believe that vaccines are effective in preventing infectious diseases (78% among Italian respondents). Furthermore, almost half of the Italian population (48% vs. 29% of Europeans) believe that vaccinations are important only for children. Therefore, the prevention of infectious diseases in groups other than children is a challenging, yet fundamental, objective that public health systems should pursue in order to promote healthy aging [5,6].

Immunization guidelines released by WHO, the United States (US) Centers for Disease Control and Prevention (CDC), the European Center for Disease Control (ECDC), and many countries include vaccination recommendations for adults and the elderly [7,8,9]. Vaccination against influenza and *Streptococcus pneumoniae* is generally recommended for at-risk adults and for the elderly; however, this definition is country-dependent ranging from ≥50 years to ≥65 years. Some countries, such as Austria, the Czech Republic, France, Greece, Italy, the United Kingdom (UK), and the United States of America (USA), also recommend vaccination against herpes zoster virus (HZV) for older adults [10]. Despite these recommendations, vaccination coverage in the elderly and at-risk groups remains unsatisfactory.

In Italy, the 2017–2019 National Immunization Plan (Piano Nazionale per la Prevenzione Vaccinale—PNPV) recommended specific vaccinations for the elderly, defined as those 65 years old and older, and at-risk adults [11]. These include influenza vaccination, pneumococcal vaccination (pneumococcal conjugate vaccine (PCV13), followed by a dose of polysaccharide vaccine (PPV23)), and the HZV vaccine. Furthermore, the PNPV set the following coverage targets: 75% minimum and 95% optimal coverage for seasonal influenza vaccination, 75% for pneumococcal vaccination in 2019, and 50% for HZV in 2019 [11]. However, these coverage targets remain unmet. According to data from the Italian Ministry of Health, influenza vaccination coverage among the elderly reached only 53.1% in the 2018/2019 season [12].

A wide range of factors may influence vaccination uptake. The magnitude of this phenomenon prompted the WHO to develop a set of tools (quantitative surveys, qualitative interviews, and related user guide) to investigate the reasons for low vaccination uptake, to improve coverage, and to evaluate interventions [13]. The general decline in coverage may be partly attributable to the so-called vaccine hesitancy, which is defined by the Strategic Advisory Group of Experts (SAGE) on WHO immunization as the tendency to delay or to refuse vaccination despite the availability of vaccination services [14]. Today, vaccine hesitancy constitutes a real threat to the health and well-being of citizens, as it undermines the effectiveness of immunization programs. It is, therefore, essential to develop skills at the local, national, and global levels in order to identify, monitor, and counteract vaccination hesitancy and to respond promptly to anti-vaccination movements in the event of disinformation or potentially adverse events [15]. According to the “Increasing Vaccination Model”, facilitating access to vaccination can remove practical obstacles to the delivery of vaccination, and it is, therefore, important in the fight against low vaccination uptake [16]. However, other factors, such as complacency (failure to perceive the risks of disease), constraints (physical and psychological barriers), calculation (searching extensively for information), and aspects pertaining to collective responsibility (willingness to protect others) also play a role in explaining vaccination behavior [17].

The uptake of vaccination must be improved to achieve the objective to reduce the morbidity, mortality, loss of quality of life, and healthcare costs caused by VPDs [18], but the available evidence indicates that vaccination coverage in older adults and at-risk groups can be considerably improved. In this light, new vaccination strategies will be needed in order to achieve this objective.

The present study aimed to identify and describe novel and unconventional strategies implemented in Italy, at the regional or national level, in order to promote vaccination among the elderly and at-risk populations, by facilitating access to vaccination in alternative settings, namely, occupational, recreational, and unconventional healthcare settings.

## 2. Materials and Methods

In this study, we conducted a literature review and a survey among regional vaccination referents and professionals in charge of vaccination departments in Italy.

The literature review was conducted by searching the MEDLINE database for the following search terms: vaccin* uptake, strategy*, campaign, policy, elderly, fragile, comorbidit*. The search was limited to articles written in English or Italian and published up to 11 March 2019. We also adopted the same search strategy in order to explore the “gray literature” and to investigate several institutional websites (WHO, WHO Europe, CDC, ECDC, Italian Ministry of Health, Italian National Institute of Health).

The inclusion criteria required that the articles should report vaccination strategies against influenza, pneumococcal diseases, and HZV, targeted at at-risk adult populations and the elderly (65 years of age or older) and carried out in unconventional settings in Italy (i.e., occupational and recreational settings or healthcare facilities other than those routinely involved in vaccinations, namely, local health authorities (LHA) and general practitioner (GP) offices).

In order to identify additional strategies, an online survey through Google Platform was launched; an e-mail invitation to participate was sent to 129 experts: directors of hygiene and public health departments and the regional vaccination referents of all Italian regions. The survey was conducted between 19 May 2019 and 16 June 2019, and two reminders were sent out in order to increase the response rate. The survey was also sent to the generic email addresses of Italian departments of prevention.

The survey, which was estimated to require about 15 minutes to complete, consisted of four sections:(1)Demographic and professional information (Italian region and department of affiliation);(2)General information on the implementation, at the regional level, of the PNPV and the introduction of an immunization information registry;(3)A section focused on the three vaccinations of interest; interviewees were asked whether, to their knowledge, any strategy that met the eligibility criteria was implemented in the last five years in their region. This section was divided into three sub-sections, one for each of the vaccinations considered: influenza, pneumococcal, and HZV;(4)A section collecting information on any strategy reported: target, setting, healthcare professionals involved, ways of inviting the target population, waiting time to receive vaccination, counseling and informative material, and final assessment. Further contacts and material (links, attached documents, regional deliberations, etc.) were also requested in order to evaluate the strategy in detail.

The survey was organized so that responses were specific to each of the vaccinations in question. Additional space was also provided in which to add information on further strategies as free text. Information on the region, setting, target population, and strategy type was assigned for each strategy reported. In addition, information on the health personnel involved, recruitment and scheduling, provision of counseling, registration of vaccination administration in the immunization information registry, and final assessment of the impact (if applicable) was collected for the strategies identified by consulting the experts.

The survey and the database generated from survey responses are available as Appendix A in native language. Hereby, we would like to clarify that this study did not need ethics committee approval as it relied on a review of the literature and on the collection of information on vaccination strategies set up at the regional/local level.

## 3. Results

The literature review yielded seven vaccination strategies, implemented in four Italian regions (Table 1): three (43%) in Lombardy, two (29%) in Liguria, one (14%) in Emilia-Romagna, and one (14%) in Trentino-South Tyrol. All strategies concerned influenza vaccination, while three of them (43%) also addressed pneumococcal and HZV vaccination (in Liguria, Emilia-Romagna, and Lombardy). Of the 654 studies identified through MEDLINE, none met the inclusion criteria. Therefore, all strategies were identified by searching the gray literature.

Regarding the consultation of experts, 19 replies (response rate: 14.7%) were eventually collected from 10 Italian regions: four (21%) from Apulia, three (16%) from Liguria, three (16%) from Lombardy, two (11%) from Calabria, two (11%) from Campania, one (5%) from Friuli Venezia Giulia, one (5%) from Lazio, one (5%) from Marche, one (5%) from Piedmont, and one (5%) from Trentino-South Tyrol. A total of 20 vaccination strategies were reported, but only 15 of these were included in the final analysis (Table 2), as three strategies had healthcare professionals as the target population of the vaccination strategy and two did not provide sufficient information. The 15 strategies were implemented in five regions and one autonomous province (AP): five (33%) in Lombardy, three (20%) in Calabria, three (20%) in Apulia, two (13%) in the AP of Bolzano, one (7%) in Friuli Venezia Giulia, and one (7%) in Marche. Of the 15 strategies included, 10 (67%) regarded influenza vaccination (three (30%) in Calabria, two (20%) in Lombardy, two (20%) in Apulia, one (10%) in Friuli Venezia Giulia, one (10%) in Marche, and one (10%) in the AP of Bolzano), four (27%) regarded pneumococcal vaccination (two (50%) in Lombardy, one (25%) in Apulia, and one (25%) in the AP of Bolzano), and one (7%) regarded HZV vaccination (in Lombardy).

The main characteristics of the strategies, stratified by type of vaccination, are reported below.

### 3.1. Influenza Vaccination Strategies

Seventeen influenza vaccination strategies met the inclusion criteria, seven from the literature review (Table 1) and 10 from the consultation of experts (Table 2). The settings of the strategies were hospital clinics (77%), rest homes (53%), healthcare residences (47%), pharmacies (29%), hospital wards (18%), prisons (18%), mobile vaccination stations (12%), accredited socio-health facilities (12%), a center for diabetes (6%), a parish (6%), a hospital facility (6%), and a stand at a fair (6%). The target populations were at-risk adults (100%), pregnant women (6%), and subjects aged ≥65 years (71%). The strategies identified from the survey of experts utilized invitation letters in 50% of cases and provided counseling in 80%. The health personnel involved in the strategies were medical doctors (100%), nurses (90%), and other health professionals (40%). A final assessment was scheduled in 70% of cases; in 86% of these, vaccination coverage was included in the evaluation criteria. Interestingly, customer satisfaction was also considered in 57% of the strategies that underwent final assessment. The results were made public in 71% of cases.

### 3.2. Pneumococcal Vaccination Strategies

Seven pneumococcal vaccination strategies met the inclusion criteria, three from the literature review (Table 1) and four from the consultation of experts (Table 2). The settings were hospital clinics (86%), rest homes (43%), hospital wards (29%), healthcare residences (29%), and an accredited socio-health and hospital facility (14%). The target populations were at-risk adults (57%) and subjects aged ≥65 years (57%). The strategies identified from the survey of experts utilized invitation letters and provided counseling in 75% of cases. The health personnel involved in the strategies were medical doctors and nurses (100% of cases), and other health professionals (25%). A final assessment was made for all the strategies, and vaccination coverage was included among the criteria evaluated in 75% of cases. In two of these latter cases (67%), the results of the assessment were made public.

### 3.3. HZV Vaccination Strategies

A total of four HZV vaccination strategies were included in the study, three from the literature review (Table 1) and one from the consultation of experts (Table 2). The settings of the strategies were hospital clinics (75%) and an accredited socio-health and hospital facility (25%). The target populations were at-risk adults (50%) and subjects aged ≥65 years (75%). The only HZV vaccination strategy identified from the consultation of experts utilized telephone invitations, included counseling, and involved medical doctors and nurses. As for its final assessment, no information was provided.

## 4. Discussion

The present study identified and described strategies implemented in Italy in unconventional settings in order to promote influenza, pneumococcal, and HZV vaccinations among at-risk adults and the elderly.

Most of the strategies identified were aimed at promoting influenza vaccination; relatively few concerned pneumococcal and/or HZV vaccinations. The main settings identified were hospital clinics, rest homes, and healthcare residences, this last setting being mainly used for influenza vaccination. In a smaller percentage of cases, pharmacies, hospital wards, prisons, mobile vaccination stations, and accredited socio-health facilities were involved. Finally, in individual strategies, a center for diabetes, a parish, a hospital facility, and a stand at a fair were the settings involved. Not all strategies considered both targets (at-risk adults and the elderly). Indeed, with regard to influenza and pneumococcal vaccinations, the target population was mainly at-risk adults, while, in the case of HZV vaccination, the main target population was subjects aged ≥65 years.

Although many strategies to promote vaccinations were introduced in recent years, and the importance of vaccination is well recognized from the public health viewpoint, many European countries are witnessing a decline in coverage, with considerable repercussions on healthcare systems, society, and the economy. Moreover, despite the recommendations issued by health institutions, vaccination coverage in the elderly and at-risk adults remains unsatisfactory. For instance, during the 2017/2018 flu season, the European Regional Office of the WHO and the ECDC issued a warning about the low uptake of influenza vaccination in Europe, particularly among people who were at high risk of developing complications, such as the elderly [19].

The declining vaccination coverages could be partly attributable to vaccination hesitancy, which is nevertheless a complex phenomenon that can have various causes. This is why it is important to work on different levels, including the removal of barriers to vaccination. In this context, novel vaccination strategies relying on new ways of recruiting patients, such as those identified in this study, would require attention.

In all the European countries, Italy among them, it is also essential to reach optimal levels of coverage for the purpose of achieving herd immunity against specific VPDs [20]. Only in this way is it possible to curb the circulation of the micro-organism responsible for the disease, thereby ensuring protection of the whole community. The impact of vaccination on the population’s health is, therefore, considerable in terms of limiting the damage caused by VPDs or their complications and reducing both direct and indirect costs [21].

In this context, vaccination is especially important for people at higher risk. Given that life expectancy is increasing and that the worldwide number of people over 60 years of age is expected to double by 2050, reaching 2.1 billion [22], and considering that the immune system weakens with increasing age, the health strategy adopted toward this aging population will have major economic and health implications in the coming years. If the potential of vaccines to reduce the morbidity, mortality, loss of quality of life, and healthcare costs caused by VPDs is to be realized, the uptake of vaccination by at-risk adults and the elderly must be improved [18]. 

To the best of our knowledge, this is the first study to collate Italian strategies implemented in unconventional settings to promote vaccination against influenza, pneumococcal diseases, and HZV; its results shed light on existing potential strategies that can help reaching desirable coverage targets. The study was based on a literature review and the consultation of experts and, although the rate of response to the survey was low, we obtained a good overview from several Italian regions. Nevertheless, the heterogeneity of strategies identified and the fact that relevant information was sometimes lacking did not allow us to make a precise comparison among strategies. In addition, because of heterogeneity and owing to the lack of quantitative measures, we were unable to assess and compare the effectiveness of the various strategies. However, the awareness of existing strategies in this field is an essential starting point in order to both identify vaccinations that still require further efforts, namely, pneumococcal and HZV vaccinations, and better design future initiatives. In particular, our research summarized some characteristics that will be useful for the implementation of further strategies to promote influenza, pneumococcal, and HZV vaccinations among at-risk adult populations and the elderly.

## 5. Conclusions

Vaccination of the elderly and at-risk adults constitutes a fundamental measure that public health systems should strengthen in order to protect these subjects from VPDs and their complications. Given that vaccination coverage rates are still far below the established targets, new strategies to promote vaccination are needed. Unconventional settings should be utilized in order to broaden the vaccination of elderly and at-risk individuals.

## Figures and Tables

**Table 1 vaccines-08-00358-t001:** Vaccination strategies identified by means of literature review.

Vaccination Strategy	Region	Setting	Target Population	Type of Strategy
Influenza vaccination	Emilia-Romagna	Cona Hospital (Ferrara);	Pregnant women	Vax Day Hospital Presides: free access vaccinations
		Vaccination clinic in “SS.ma Annunziata” hospital, Cento;		
		“Del Delta” Hospital (Lagosanto, Ferrara);		
		Argenta Hospital (Ferrara)		
	Liguria	Hygiene Unit clinics in “San Martino” hospital Pharmacies	≥65 years, adults with risk conditions	Campaign promoted by Liguria Region and A.Li.Sa., in collaboration with five Health Authorities and Ligurian Hospitals
	Liguria	Vaccination clinic in “Villa Scassi” hospital (Genova)	Fragile hospitalized patients, pregnant women, and those who live with immunosuppressed subjects	“Ospivax, the vaccinating hospital” Project
	Lombardy	Dedicated clinics of some specialist departments	≥65 years, adults with risk conditions, and pregnant women	Influenza vaccination campaign
	Lombardy	Accredited socio-health and hospital facilities	≥65 years, adults with risk conditions, pregnant women, those working on livestock farms, healthcare personnel, and public services personnel	Influenza vaccination campaign
	Lombardy	“San Camillo” rest home and other social and health facilities	≥65 years, adults with risk conditions	Influenza vaccination campaign
	Trentino-South Tyrol	Stand at the Autumn Fair	≥65 years, adults with risk conditions	Influenza vaccination campaign
Pneumococcal and HZV vaccinations	Emilia-Romagna	Cona Hospital (Ferrara);	People aged 65 and 66	Vax Day Hospital Presides: free access vaccinations
		Vaccination clinic in “SS.ma Annunziata” hospital, Cento;		
		“Del Delta” Hospital (Lagosanto, Ferrara);		
		Argenta Hospital (Ferrara)		
	Liguria	Vaccination clinic in “Villa Scassi” hospital (Genova)	Fragile hospitalized patients and those who live with immunosuppressed subjects	“Ospivax, the vaccinating hospital” Project
	Lombardy	Accredited socio-health and hospital facilities	≥65 years	Free pneumococcal and HZV vaccination

Note: HZV: Herpes Zoster Virus; A.Li.Sa: Azienda Ligure Sanitaria della Regione Liguria.

**Table 2 vaccines-08-00358-t002:** Vaccination strategies identified by consulting experts.

Vaccination Strategy	Region	Province	Target Population	Setting	Access to Vaccination	Invitation to Vaccination Initiative	Counseling (Supporting Information Material)	Registration in the Vaccination Registry?	Strategy Assessment	Evaluation Parameters of the Strategy	Are the Results Public?
Influenza vaccination	Calabria	Crotone	≥65 years, adults with risk conditions	Hospital clinic, health care residence, pharmacy, prison house, rest home	Paper booking	Invitation letter	No	No	No	N.A.	N.A.
	Calabria	Cosenza	≥65 years, adults with risk conditions	Hospital clinic, health care residence, pharmacy, prison house, rest home, parish	Paper, telephone, and online booking	Invitation letter, text message	Yes (yes)	Yes	Yes	% uptake, vaccination coverage, customer satisfaction	Yes
	Calabria	Cosenza	≥65 years, adults with risk conditions	Mobile vaccination station, hospital clinic, health care residence, pharmacy, prison house, rest home	Paper, telephone, and online booking	Invitation letter, text message	Yes (yes)	Yes	Yes	% uptake, vaccination coverage, costs, customer satisfaction	Yes
	Friuli Venezia Giulia	Pordenone	≥65 years, adults with risk conditions	Hospital clinic, health care residence, pharmacy, rest home	Telephone and online booking	By telephone contact, by email	Yes (yes)	Yes	Yes	% uptake, vaccination coverage, costs	Yes
	Lombardy	Milan	≥65 years, adults with risk conditions	Hospital ward, hospital clinic, health care residence, rest home	N.R.	Media, posters, website	Yes (yes)	Yes	Yes	% uptake, vaccination coverage, costs	No
	Lombardy	Sondrio	≥65 years, adults with risk conditions	Hospital clinic, health care residence, rest home	Telephone booking	Invitation letter	Yes (no)	Yes	N.R.	N.A.	N.A.
	Marche	Ancona	Adults with risk conditions	Mobile vaccination station, center for diabetes	N.R.	N.R.	Yes (yes)	No	No	N.A.	N.A.
	Apulia	Foggia	Adults with risk conditions	Hospital ward, hospital clinic, health care residence, rest home	Telephone and online booking	Invitation letter, by email	Yes (no)	Yes	Yes	% uptake, vaccination coverage, customer satisfaction, costs	Yes
	Apulia	Bari	Pregnant women	Hospital clinic	Telephone booking	By telephone contact	Yes (yes)	Yes	Yes	% uptake, customer satisfaction, adverse event with call after 48/72 h	No
	Trentino-South Tyrol	Bolzano	≥65 years, adults with risk conditions	Hospital ward, hospital clinic, rest home	Paper, telephone, and online booking	N.R.	No	Yes	Yes	% uptake, vaccination coverage	Yes
Pneumococcal vaccination	Lombardy	Milan	Adults with risk conditions	Hospital ward, hospital clinic	N.R.	N.R.	Yes (yes)	Yes	Yes	Vaccination coverage, costs	No
	Lombardy	Sondrio	≥65 years, adults with risk conditions	Hospital clinic, health care residence, rest home	Telephone booking	Invitation letter, by telephone contact	Yes (no)	Yes	N.R	N.A	N.A
	Apulia	Foggia	Adults with risk conditions	Hospital ward, hospital clinic, health care residence, rest home	Telephone and online booking	Invitation letter, by e-mail	Yes (yes)	Yes	Yes	% uptake, vaccination coverage, costs	Yes
	Trentino-South Tyrol	Bolzano	≥65 years	Hospital clinic, rest home	Paper, telephone, and online booking	Invitation letter	No	Yes	Yes	% uptake, vaccination coverage	Yes
HZV vaccination	Lombardy	Sondrio	≥65 years, adults with risk conditions	Hospital clinic	Paper and telephone booking	By telephone contact	Yes (no)	Yes	N.R.	N.A.	N.A.

Note: N.R.: not reported; N.A.: not applicable; HZV: Herpes Zoster Virus.

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
