# Peer review of "Strategies to Improve Vaccination among At-Risk Adults and the Elderly in Italy"

_vaccines, 2020, doi:10.3390/vaccines8030358_

Round 1

Reviewer 1 Report

The paper is difficult to read since it is a mixture of a literature search and opinions of an expert panel.

The aim of the article is to compare different strategies for increasing the vaccine uptake of 3 vaccine (influenza, pneumococcal vaccines and HZV vaccine) in the elderly and other immunocompromised individuals There is very little written about strategies and no comparison at all of different strategies (personal contact with individuals at risk, advertising etc). Thus, it does not help the reader to judge which strategies are the best to increase vaccine uptake in his/her region.

A detail: The title has a strange mixture of capital and small letters.

I do not think the manuscript is of sufficient quality to be published.

Author Response

We thank the Reviewer for the general comment and his/her suggestions that have given us the possibility to improve the quality of the manuscript. Hereafter, point-by-point answers have been reported.

  1. The paper is difficult to read since it is a mixture of a literature search and opinions of an expert panel.

Thanks for your observation. We are aware of difficulties that you could meet in reading the paper. Nevertheless, the fact that we put together two different sources of information was reasoned by the need for being as exhaustive as possible in describing current strategies to improve vaccination coverage set up in unconventional settings.

  1. The aim of the article is to compare different strategies for increasing the vaccine uptake of 3 vaccine (influenza, pneumococcal vaccines and HZV vaccine) in the elderly and other immunocompromised individuals. There is very little written about strategies and no comparison at all of different strategies (personal contact with individuals at risk, advertising etc). Thus, it does not help the reader to judge which strategies are the best to increase vaccine uptake in his/her region.

Thanks for your observation. Actually, the aim of the study was not to compare strategies but “to identify novel and unconventional strategies implemented in Italy, at the regional or national level, in order to improve vaccination coverage among elderly and at-risk populations, by facilitating access to vaccination in alternative settings, namely occupational, recreational and unconventional healthcare settings”. Hence the aim was merely descriptive and we have included “and describe” after “identify” to better address this point. Furthermore, we have amended the final part of the discussion in order to better describe the added value of our work in the field of vaccination strategies. We obviously agree with you on the importance of identifying best strategies but, besides being out of our scope, the comparison was also hampered by the fact that a final evaluation was not performed in all cases and that, among the strategies eventually assessed, vaccination coverage was not always addressed and results were not always public.

  1. A detail: The title has a strange mixture of capital and small letters.

Thanks for your observation. BRAVE is the acronym of the project and the mixture of capital and small letters has been used in reference to it. Nevertheless, in order to make the title clearer, we have rephrased it.

  1. I do not think the manuscript is of sufficient quality to be published.

We respect your final judgment and we hope that, following our revision, the manuscript could be now considered acceptable.

Reviewer 2 Report

Please see the document attached. 

Author Response

We thank the Reviewer for the general comment and his/her suggestions that have given us the possibility to improve the quality of the manuscript. Hereafter, point-by-point answers have been reported.

  1. Line 21: Are there any that don’t address these populations? Seems odd that they wouldn’t address this group. Could be reworded to say: …”for adults and the elderly.” Just a bit smoother.

Thanks for your suggestion. We have modified the text accordingly.

  1. Line 34: Would recommend using “influenza” rather than shortened “flu”.

Thanks for your suggestion. We have modified the text accordingly.

  1. Line 78-80: I imagine that you also included a search term to exclude strategies outside of Italy. Can you provide more detail if this is the case? Also, were there any terms used to identify your unconventional settings?

Thanks for your request. We did not use search terms to exclude strategies set up in other countries but we considered that point as an inclusion criterion. Similarly, as unconventional settings are by definition not commonly used to provide vaccination, we did not include any search term related to them but we used them as a selection criterion.

  1. Line 87-88: Did you have an age that you searched?

Thanks for your request. According to the most common definition of elderly, we considered strategies targeting people aged 65 years old or over and we have specified it in the text.

  1. Line 110: This is a bit confusing. Is this an accurate reword? “The survey was organized so that responses were specific to each of the vaccinations in question.”

Thanks for your suggestion. We have modified the text accordingly.

  1. Line 113: I would call this “scheduling” but that may be just my preference

Thanks for your suggestion. We have modified the text accordingly.

  1. Line 114: I’m sorry. I don’t know what this means. For other readers, consider providing more detail.

Thanks for your observation. We have replaced vaccination registry with immunization information registry and we have specified “of vaccination administration” after registration in order to make it clear that we referred to the system in which data about vaccine doses administration of each individual is recorded.

  1. Line 119: I don’t think these percentages are needed.

Thank you. We agree with your comment and we have removed 100%.

  1. Line 120-121: Does this mean that no strategies were identified with your Medline lit search? If so, should include sentence addressing this. Could say something to the effect of *** trials were identified through Medline search, however zero met inclusion criteria.

Thank you for your observation. Of the 654 studies identified through MEDLINE none met inclusion criteria and we have included this sentence in the text for the sake of clarity.

  1. Line 136: I like how you organized alphabetically by region.

Thank you.

  1. Line 142-143: Like education for healthcare providers on the need for vaccination?

Thanks for your question. Strategies targeting healthcare professionals that were excluded were those aimed at increasing healthcare professionals’ vaccination coverage. Indeed, these articles were about strategies that had healthcare professionals as target population.

  1. Line 183-184: Percentages?

Thanks for your question. Percentages were no reported as this kind of information was available only for strategies identified through the experts’ consultation and, in regard to HZV, only one strategy was found. In order to be more clear with respect to this point, the information on health personnel involved was moved after the description of ways for patients’ recruitment and counselling in all sections 3.1, 3.2 and 3.3.  

  1. Line 187: Keep discussion 188-436. Consider then making the argument that because of low vaccination coverage, need to try new things! For example new ways of recruiting patients for vaccination and removing barriers to vaccination. Can then pull current content from 456-467 into argument as to why these groups of patients are so important in these efforts.

Thanks for this suggestion. We have included the following sentence in the discussion “The declining vaccination coverages could be partly attributable to vaccination hesitancy, which is anyway a complex phenomenon that can have various causes. This is why it is important to work on different levels, including the removal of barriers to vaccination. In this context, novel vaccination strategies relied on new ways of recruiting patients, such as those identified in this study, would require attention”. On the other hand, we have not been able to identify the content you refer speaking about lines 456-467 (as these lines are not present at all in the text).

  1. Line 200-201: Recognized by who? Seems somewhat contradictory to the data you have above regarding survey of citizens and their perception of vaccine importance.

Thanks for the warning. We have rephrased the sentence as follows “the importance of vaccination is well recognized from the public health viewpoint”.

  1. Line 207-220: These seem like thoughts that you should move into the introduction. Specifically would recommend new paragraph following paragraph where you review the PNPV data. Then, once introduced in the introduction section of the article, you can refer back to them here with a few sentences.
  2. Line 221-223: Recommend adding to introduction paragraph currently at line 77.
  3. Line 223-226: Same as above, consider moving to introduction to set the stage as to why vaccine uptake in these populations is likely so low.

Thanks for your suggestion. We have moved these parts in the introduction and we have added few lines concerning vaccine hesitancy and the importance of removing barriers in the discussion.

Reviewer 3 Report

This article reviews vaccine strategies in unconventional settings in Italy for protection of elderly and high-risk populations. It includes a survey of public health experts on the topic as well.

A major shortfall of the study is identified on lines 253-254: “In  addition,  owing  to  the  lack  of  quantitative  measures,  we  were  unable  to compare the effectiveness of the various strategies.” Compiling the vaccine strategies in unconventional settings, without identifying their efficacy, is of minimal utility in informing public health vaccination policy efforts. Because of this, it may be preferable to re-write this article as a brief communication addressing the need for an efficacy study rather than presenting it as an original research article.

The data availability policy of Vaccines requires, “In order to maintain the integrity, transparency and reproducibility of research records, authors must make their experimental and research data openly available either by depositing into data repositories or by publishing the data and files as supplementary information in this journal.” Please either make the database generated from survey responses available as a supplemental data file or available in a public repository, and reference this in the methods section of the manuscript. I necessary to interpret the database, please also include the text of the survey instrument.

Table 1 would be improved by including references to the literature reporting each vaccination strategy.

Minor comments

Correct the capitalization in the title

Please use the full “Herpes Zoster Virus (HZV)” for the first mention in the abstract

Line 28: delete “anti-“

Line 57-59: this is directly repeated from the abstract

Line 38, 64 remove “however,” it is redundant

Line 79-81: awkward/passive voice

Line 98: insert a comma after “pneumococcal disease”

Line 182: Please include citations for the published studies.

Section 3.2 & 3.3: Please combine each section into paragraphs rather than single indented lines.

Paragraph Lines 219-228: This discusses vaccine hesitancy as a discussion point; however, none of the data seems to concern vaccine hesitancy. This paragraph does not relate well to the rest of the manuscript.

Line 250: What was the survey response rate?

Author Response

We thank the Reviewer for the general comment and his/her suggestions that have given us the possibility to improve the quality of the manuscript. Hereafter, point-by-point answers have been reported.

This article reviews vaccine strategies in unconventional settings in Italy for protection of elderly and high-risk populations. It includes a survey of public health experts on the topic as well.

  1. A major shortfall of the study is identified on lines 253-254: “In addition, owing o the lack of quantitative measures, we were unable to compare the effectiveness of the various strategies.” Compiling the vaccine strategies in unconventional settings, without identifying their efficacy, is of minimal utility in informing public health vaccination policy efforts. Because of this, it may be preferable to re-write this article as a brief communication addressing the need for an efficacy study rather than presenting it as an original research article.

Thank you for your comment. We agree with you that it would be important to assess the efficacy of identified strategies. Nevertheless, as also described in results, not all the strategies were eventually evaluated and this prevented us to make an assessment of their effectiveness. Furthermore, the topic might be considered emerging as academic evidence on strategies implemented in unconventional settings is still scant. This justifies why we decided to submit a research articles, to better describe how we looked for these strategies and which characteristics they had. Eventually, we have also amended the final part of the discussion in order to better describe the added value of our work in the field of vaccination strategies.

  1. The data availability policy of Vaccines requires, “In order to maintain the integrity, transparency and reproducibility of research records, authors must make their experimental and research data openly available either by depositing into data repositories or by publishing the data and files as supplementary information in this journal.” Please either make the database generated from survey responses available as a supplemental data file or available in a public repository, and reference this in the methods section of the manuscript. I necessary to interpret the database, please also include the text of the survey instrument.

Thanks for your request. The survey and the database have been now included as supplementary material.

  1. Table 1 would be improved by including references to the literature reporting each vaccination strategy.

Thanks for your suggestion. Nevertheless, we have not included the references in table 1 because they mainly refer to websites links that are no longer accessible (e.g. webpages of healthcare structures dedicated to the disclosure of their local initiatives). In addition, we have tried to standardize the format of table 1 with table 2.

Minor comments

  1. Correct the capitalization in the title

Thanks for your observation. BRAVE is the acronym of the project and the mixture of capital and small letters have been used in reference to it. Nevertheless, in order to make the title clearer, we have modified it.

  1. Please use the full “Herpes Zoster Virus (HZV)” for the first mention in the abstract

Thanks for the warning. We have reported the full “Herpes Zoster Virus (HZV)” in the abstract.

  1. Line 28: delete “anti-“

Thanks for the warning. We have deleted “anti”.

  1. Line 57-59: this is directly repeated from the abstract

Thanks for the warning. The sentence has been rephrased in order to differentiate it from the one in the abstract.

  1. Line 38, 64 remove “however,” it is redundant

Thanks for the suggestion. We have deleted “however” from the two lines.

  1. Line 79-81: awkward/passive voice

Thank you for the suggestion. The sentence has been rephrased as follows “The uptake of vaccination must be improved to achieve the objective to reduce the morbidity, mortality, loss of quality of life and healthcare costs caused by VPDs”

  1. Line 98: insert a comma after “pneumococcal disease”

Thanks for the warning. The comma has been included.

  1. Line 182: Please include citations for the published studies.

Thanks for the observation. We have modified the sentence as follows “In 2 of these latter cases (67%) the results of the assessment were made public” in order to be coherent with the previous section and to make it clear that results were released not in the form of a paper.

  1. Section 3.2 & 3.3: Please combine each section into paragraphs rather than single indented lines.

Thanks for the suggestion. We have combined sentences into a single paragraph in Sections 3.1, 3.2 and 3.3.

  1. Paragraph Lines 219-228: This discusses vaccine hesitancy as a discussion point; however, none of the data seems to concern vaccine hesitancy. This paragraph does not relate well to the rest of the manuscript.

Thanks for your suggestion. Following also a comment from another Reviewer we have moved this part into the introduction. We have just left a reference to vaccine hesitancy in order to justify the interest in the implementation of novel strategies to provide vaccinations.

  1. Line 250: What was the survey response rate?

Thanks for your request. We have reported the response rate in the results section.

Round 2

Reviewer 1 Report

The manuscript is considerably improved and the subject is important. Vaccination of the elderly and immunocopromised is much, much too low both in Europé and North America.

Minor comments for improvement:

  1. Title: I still do not understand why the authors have such a strange mixture of capital and small letters. Usually the first letter in a sentence has a capital letter and the rest are small. As they write it now pRActices is completely wrong.
  2. Line 101. Should be vaccines, not vaccine*
  3. Line 223-224. Vaccination of the elderly does not induce herd immunity, because the elderly are seldom spredaers of the three diseases discussed. Influenza changes type every few years, so it is not possible to induce long lasting herd immunity. If you want to achieve herd immunity against the currently ongoing type you should vaccinate children and adolescents, who are the important spreaders of the virus. For instance in the US they recommend vaccination of all children up to 8 years, which is very costly and difficult to perform so the vaccination rate in children is low. As for pneumocci, the elderly get more severely than other age groups, bur the reservoir of the organsim are pre-school children who spread it to the elderly and immunocompromised. Zoster is always a reactivation of a previous varicella infection. Herpes zoster has little contagiousness. Vaccination of the elderly and immunocompromised helps the individual but has little or no effect on herd immunity. The only way of getting herd immunity against herpes zoster is to vaccinate all infants against varicella, Then they do probably not get zoster at old age or when they get immunocompromising diseases. But how that will work we will not know in several decades.

Author Response

The manuscript is considerably improved and the subject is important. Vaccination of the elderly and immunocopromised is much, much too low both in Europé and North America.

We thank the Reviewer for the general comment and his/her suggestions that have given us the possibility to improve the quality of the manuscript. Hereafter, point-by-point answers have been reported.

Minor comments for improvement:

Title: I still do not understand why the authors have such a strange mixture of capital and small letters. Usually the first letter in a sentence has a capital letter and the rest are small. As they write it now pRActices is completely wrong.

Thanks for the warning. We accept your proposal and we have modified the title of the paper as follows “Strategies to improve vaccination among at-risk adults and the elderly in Italy”

Line 101. Should be vaccines, not vaccine*

Thanks for the warning. We have actually used “vaccin*” in order to catch plural and singular nouns.

Line 223-224. Vaccination of the elderly does not induce herd immunity, because the elderly are seldom spredaers of the three diseases discussed. Influenza changes type every few years, so it is not possible to induce long lasting herd immunity. If you want to achieve herd immunity against the currently ongoing type you should vaccinate children and adolescents, who are the important spreaders of the virus. For instance in the US they recommend vaccination of all children up to 8 years, which is very costly and difficult to perform so the vaccination rate in children is low. As for pneumocci, the elderly get more severely than other age groups, bur the reservoir of the organsim are pre-school children who spread it to the elderly and immunocompromised. Zoster is always a reactivation of a previous varicella infection. Herpes zoster has little contagiousness. Vaccination of the elderly and immunocompromised helps the individual but has little or no effect on herd immunity. The only way of getting herd immunity against herpes zoster is to vaccinate all infants against varicella, Then they do probably not get zoster at old age or when they get immunocompromising diseases. But how that will work we will not know in several decades.

Thanks for your important observation. We agree with you and we have modified the sentence as follows “also for the purpose of achieving herd immunity against specific VPDs”. Please bear in mind that the sentence is included in a generic statement on the importance of all vaccinations.